# NEIGHBOURHOOD DISTILLATION:
# ON THE BENEFITS OF NON END-TO-END DISTILLATION

## ABSTRACT

End-to-end training with back propagation is the standard method for training deep neural networks. However, as networks become deeper and bigger, end-to-end training becomes more challenging: highly non-convex models gets stuck easily in local optima, gradients signals are prone to vanish or explode during back-propagation, training requires computational resources and time. In this work, we propose to break away from the end-to-end paradigm in the context of Knowledge Distillation. Instead of distilling a model end-to-end, we propose to split it into smaller sub-networks - also called neighbourhoods - that are then trained independently. We empirically show that distilling networks in a non end-to-end fashion can be beneficial in a diverse range of use cases. First, we show that it speeds up Knowledge Distillation by exploiting parallelism and training on smaller networks. Second, we show that independently distilled neighbourhoods may be efficiently re-used for Neural Architecture Search. Finally, because smaller networks model simpler functions, we show that they are easier to train with synthetic data than their deeper counterparts.

## 1 INTRODUCTION

As Deep Neural Networks improve on challenging tasks, they also become deeper and bigger. Image classification convolutional neural networks grew from 5 layers in LeNet (LeCun et al., 1998) to more than a 100 in the latest ResNet models (He et al., 2016). However, as models grow in size, training by back propagating gradients through the entire network becomes more challenging and computationally expensive. Convergence in a highly non-convex space can be slow and requires the development of sophisticated optimizers to escape local optima (Kingma & Ba, 2014). Gradients vanish or explode as they get passed through an increasing number of layers. Very deep neural networks that are trained end-to-end also require accelerators, and time to train to completion.

Our work seeks to overcome the limitations of training very deep networks by breaking away from the end-to-end training paradigm. We address the procedure of distilling knowledge from a teacher model and propose to break a deep architecture into smaller components which are distilled independently. There are multiple benefits to working on small neighbourhoods as compared to full models: training a neighbourhood takes significantly less compute than a larger model; during training, gradients in a neighbourhood only back-propagate through a small number of layers making it unlikely that they will suffer from vanishing or exploding gradients. By breaking a model into smaller neighbourhoods, training can be done in parallel, significantly reducing wall-time for training as well as enabling training on CPUs which are cheaper than custom accelerators but are seldom used in Deep Learning as they are too slow for larger models.

Supervision to train the components is provided by a pre-trained teacher architecture, as is commonly used in Knowledge Distillation (Hinton et al., 2015), a popular model compression technique that encourages a student architecture to reproduce the outputs of the teacher. For this reason, we call our method Neighbourhood Distillation. In this paper, we explore the idea of Neighbourhood Distillation on a number of different applications, demonstrate its benefits, and advocate for more research into non end-to-end training.

**Contributions**

- We provide empirical evidence of the thresholding effect, a phenomenon that highlights deep neural networks' resilience to local perturbations of their weights. This observation motivates the idea of Neighbourhood Distillation.

- We show that Neighbourhood Distillation is up to 4x faster than Knowledge Distillation while producing models of the same quality. We demonstrate this on model compression and sparsification.

- Then, we show that neighbourhoods trained independently can be used in a search algorithm that efficiently explores an exponential number of possibilities to find an optimal student architecture.

- Finally, we show applications of Neighbourhood Distillation to zero-data settings. Shallow neighbourhoods model less complex functions which we can distill using only Gaussian noise as a training input.

## 2 RELATED WORK

**Non end-to-end training**  Before the democratization of deep learning, machine learning methods relied on multi-stage pipelines. For example, the face detection algorithm designed by Viola & Jones (2001) is a multi-stage pipeline relying first on handcrafted feature extraction and then on a classifier trained to detect faces from the features. Then came the idea of directly learning classification from the input image, leaving the model to learn all parts of the pipeline through a series of hidden layers (LeCun et al., 1998; Fukushima & Miyake, 1982) that could be trained with end-to-end with gradient back-propagation (Rumelhart et al., 1986) or layerwise training (Vincent et al., 2008). End-to -end deep learning gained traction with the success of the AlexNet model (Krizhevsky et al., 2012) in image classification. It is now the main component of various state-of-the art approaches in object detection (Redmon et al., 2016; Ren et al., 2015), image segmentation (He et al., 2017), speech processing (Senior et al., 2012), machine translation (Seo et al., 2016; Vaswani et al., 2017).

However, gradient-based end-to-end learning comes with a cost. Highly non-convex losses are harder to optimize; models of bigger sizes also require more data to fully train; they suffer from vanishing and exploding gradients (Hochreiter, 1998; Pascanu et al., 2012).

Approaches to overcome these issues can be broken down into three categories. First, several methods have been introduced to ease the training of deep models, such as residual connections (He et al., 2016), gated recurrent units (Cho et al., 2014), normalization layers (Ioffe & Szegedy, 2015; Ba et al., 2016; Salimans & Kingma, 2016), and more powerful optimizers (Kingma & Ba, 2014; Hinton et al.; Duchi et al., 2011). Second, engineering best practices have adapted to rise to the challenges raised by deep learning: pre-trained models trained on large-datasets can be reused for transfer learning, only requiring the fine-tuning of a portion of the model for specific tasks (Devlin et al., 2018; Dahl et al., 2011). Distributed training (Krizhevsky et al., 2012; Dean et al., 2012) and custom hardware accelerators (Jouppi et al., 2017) were also crucial in accelerating training.

The last category, which our work falls into, investigates non end-to-end training methods for deep neural networks. One class of non end-to-end learning method relies on splitting a deep network into gradient-isolated modules trained with local objectives (Löwe et al., 2019; Nøkland & Eidnes, 2019). Layerwise training (Belilovsky et al., 2018; Huang et al., 2017) also divides the target network into modules that are sequentially trained in a bottom-up approach. Difference Target Propagation (Lee et al., 2015) seeks to optimize each layer to output activations close to a given target value. These values are computed by propagating inverses from downstream layers while ours are provided by a pre-trained teacher model. All of these approaches also differ from ours as modules still depend on each other, while our neighbourhoods are distilled independently.

**Knowledge Distillation**  Our work specifically draws from Knowledge Distillation (Hinton et al., 2015), a general-purpose model compression method that has been successfully applied to vision (Crowley et al., 2018) and language problems (Hahn & Choi, 2019). Knowledge Distillation transfers knowledge from a teacher in the form of its predicted soft logits. Various variations have been developed to improve distillation. One direction is to transfer additional knowledge in the form of intermediate activations (Romero et al., 2014; Aguilar et al., 2019; Zhang et al., 2017), attention maps (Zagoruyko & Komodakis, 2016), weight projections (Hyun Lee et al., 2018) or layer

interactions (Yim et al., 2017). Other methods also seek to directly address the capacity gap between a teacher and student (Cho & Hariharan, 2019) by distilling from a series of intermediate teachers (Mirzadeh et al., 2019; Jin et al., 2019). These methods all distill the student end-to-end.

**Neural Architecture Search**   Recent papers study how to combine Knowledge Distillation with Neural Architecture Search methods, which automate the design process by exploring a given search space (Liu et al., 2018; Pham et al., 2018; Liu et al., 2017; Tan & Le, 2019; Zoph et al., 2017). These methods have successfully been applied to find better suited students for a given teacher (Kang et al., 2019; Liu et al., 2020). Closely related to our work, Li et al. (2020) divide a supernet into blocks and use Knowledge Distillation to train it. However, they only focus on extracting the architectural knowledge from the teacher, ignoring the parameters learned during the search process.

## 3   Thresholding Effect

Due to their size, modern neural networks are usually overparameterized. Their learned representations are redundant and recent empirical studies conducted on ResNets (Zhang et al., 2019; Veit et al., 2016) showed that it is possible to drop or reset layers in a trained network without hurting their performance. We hypothesize further that less drastic modifications in a network, such as replacing part or all of their sub-components by imperfect approximations, will not result in dramatic error accumulation. In the following section, we provide empirical evidence of an interesting property that supports this: sub-components of a trained model may be perturbed without damaging the model's accuracy, as long as individual local errors remain under a certain threshold. We call this phenomenon the *thresholding effect*.

First, we introduce the notion of neighbourhood that will be used throughout the rest of the paper. Deep neural networks are built by stacking a succession of blocks of operations such as convolutions and non-linear layers. We express this by defining a network $\mathcal{T}$ as a composition of $n$ sub-networks:

$$\forall i \in \{1 \dots n\}, \mathcal{T}_i : \mathbb{R}^{F_i} \to \mathbb{R}^{F_{i+1}}$$
$$\mathcal{T} = \mathcal{T}_n \circ \mathcal{T}_{n-1} \circ \cdots \circ \mathcal{T}_1 \tag{1}$$

We call neighbourhood any portion of the network that is delimited by one sub-network $\mathcal{T}_i$. This neighbourhood represents an arbitrary logical construction block in the network and may be replaced by variants of its architecture that have the same input and output shapes. For any given netwrok, one can define multiple ways to break it up into neighbourhoods.

The question we set out to answer is the following. Imagine we want to replace part or all the neighbourhoods by imperfect approximations, how good do these approximations need to be to prevent a drastic loss of performance in the modified model?

We consider different pre-trained models and estimate how their accuracy is impacted by perturbations of the network's intermediate features. To do so, we perturb each neighbourhood by artificially introducing some gaussian noise of amplitude $\epsilon$ to each activation output.

$$\mathcal{S}_i = \mathcal{T}_i + \delta \tag{2}$$
$$\delta \sim \mathcal{N}(0, \epsilon^2)$$

The students are then composed into a full network $\mathcal{S} = \mathcal{S}_n \circ \mathcal{S}_{n-1} \cdots \circ \mathcal{S}_1$ which we evaluate.

On CIFAR-10 (Krizhevsky et al., 2009), we train a ResNetV1-20 (He et al., 2016) model and define a neighbourhood as one bottleneck block which consists of a two-layer convolutional network and a skip connection. We reiterate a similar experiment on a large-scale dataset. On ImageNet (Deng et al., 2009), we train several EfficientNet (Tan & Le, 2019) models and define a neighbourhood as one mobile inverted bottleneck block.

Figure 1 shows how errors of different amplitudes accumulate across networks when replacing some or all neighbourhoods by approximations. In particular, we consistently witness a thresholding effect in all networks. When the amplitude $\epsilon$ of the noise is small enough, the final accuracy of the network is not impacted by accumulated perturbations. This threshold appears to depend on the number of neighbourhoods.

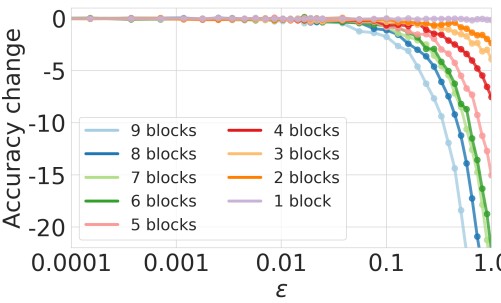
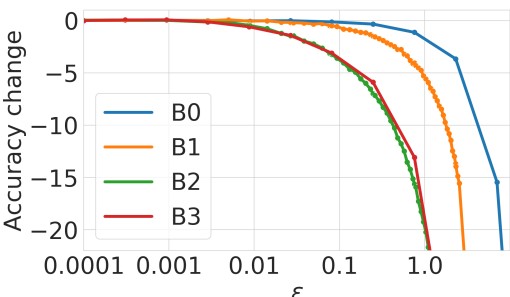

(a) ResNetV1-20 with $91.6\%$ accuracy on CIFAR-10. Each line represents a different number of perturbed blocks.

(b) EfficientNet models trained on ImageNet. Each line represents a different EfficientNet model for which all neighbourhoods have been perturbed.

Figure 1: We perturb the intermediate outputs at regular locations in the network using a gaussian noise of amplitude $\epsilon$ and measure the effect of these perturbations on the accuracy of the model. We show that accuracy remains stable (accuracy change close to $0$) as long as $\epsilon$ remains under a certain threshold. The threshold differs between network architectures and number of perturbed neighbourhoods. Note that for all models, even when perturbing all neighborhoods, there is still a range for which there is virtually no loss in accuracy.

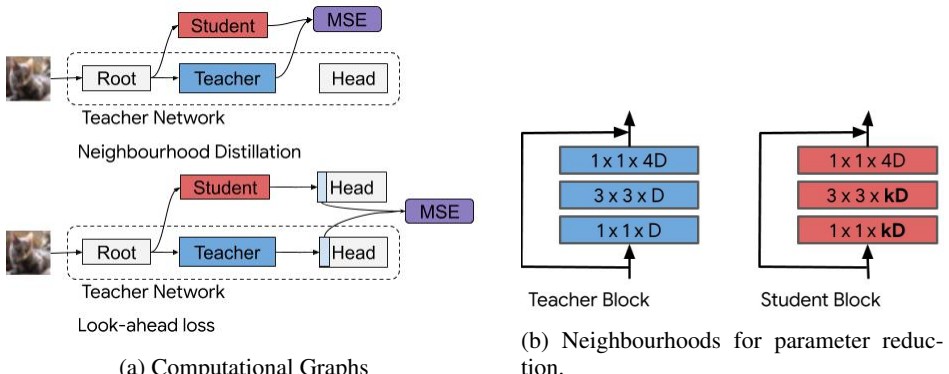

(a) Computational Graphs

(b) Neighbourhoods for parameter reduction.

Figure 2: (a) Computation graphs for Neighbourhood Distillation. Top: the teacher and the student neighbourhoods receive activations from the root of the teacher network. The student neighbourhood is trained to reproduce the output of the teacher. Bottom: the teacher and student outputs are propagated to the head of the teacher network and additional activations between teacher and student networks are compared. The look-ahead loss gives an additional training signal for the student to reproduce the teacher. (b) Example of teacher and student neighbourhoods.

Our experiments on the thresholding effect show that it is possible to locally replace sub-components of a model without hurting the performance of the reconstructed model. This observation is what motivates Neighbourhood Distillation: neighbourhoods trained to approximate their teacher outputs can be used to reconstruct student networks with no or limited accuracy drop.

In the appendix, we also present preliminary results on understanding the thresholding effect and show how regularizing the teacher network can impact the threshold.

# 4 NEIGHBOURHOOD DISTILLATION

In order to train a new model in a non end-to-end fashion, we leverage the representation of a network as a stack of neighbourhoods. Supervision to train each component independently is provided by a pre-trained teacher architecture which is also divided into small neighbourhoods. Each student neighbourhood $\mathcal{S}_i$ is trained to match the outputs of its teacher by minimizing the mean-square error between their produced features. The training inputs are obtained by forward-propagation of the

Table 1: Neighbourhood Distillation results on CIFAR-10 for different values of multiplier $k$. Our Neighbourhood Distillation method reaches the same accuracy as Knowledge Distillation (KD) and performs better than retraining from scratch (CE). As $k$ decreases, fine-tuning becomes critical in order to recover good accuracy. Our method is $2.3\times$ faster than Knowledge Distillation on GPU.

| MULTIPLIER $k$ | # PARAMS | CE | KD | ND + FT | ND |
|---|---|---|---|---|---|
| 1 | 269K | 91.6 | **91.9** | 91.6 | 91.6 |
| 0.9 | 239K | 91.3 | **91.7** | **91.7** | 91.3 |
| 0.8 | 213K | 90.8 | 91.5 | **91.7** | 90.7 |
| 0.75 | 202K | 90.9 | 91.6 | **91.7** | 87.6 |
| 0.6 | 160K | 90.5 | 90.6 | **91.1** | 90.8 |
| 0.5 | 136K | 90.2 | 91.0 | **91.0** | 88.2 |
| 0.4 | 105K | 89.3 | 89.7 | **90.0** | 83.5 |
| GPU TIME(H) | - | - | 3.2 | 1.4 | 0.32 |
| SPEED-UP | - | - | $1\times$ | $2.3\times$ | $10\times$ |

original training images through the first part of the teacher network $\mathcal{T}_{i-1} \circ \cdots \circ \mathcal{T}_1$. The computational graph for Neighbourhood Distillation is shown in Figure 2a. These activation maps can be pre-computed and stored before training. This makes each student neighbourhood extremely fast to train, as one does not need to compute activations through a big model in each training step.

Different student neighbourhoods can be distilled *independently from each other* and composed into a final student model $\mathcal{S} = \mathcal{S}_n \circ \cdots \circ \mathcal{S}_1$. The final student model is then fine-tuned by optimizing the Knowledge Distillation loss from Equation 3.

$$\mathcal{L}_{KD}(\mathbf{x}, \mathbf{y}) = \mathcal{L}_{CE}(\hat{\mathbf{y}}_\tau^\mathcal{T}, \hat{\mathbf{y}}_\tau^\mathcal{S}) + \lambda \mathcal{L}_{CE}(\mathbf{y}, \hat{\mathbf{y}}_1^\mathcal{S}) \tag{3}$$
$$\hat{\mathbf{y}}_\tau^\mathcal{T} = softmax(\frac{\mathcal{T}(\mathbf{x})}{\tau})$$

where $\mathcal{L}_{CE}$ is the standard cross-entropy loss.

While the idea of minimizing the mean-square error between output features is similar to Hint-Training (Romero et al., 2014), our method is more general as we do not limit ourselves to distill one prefix sub-network. Neighbourhood Distillation also provides more flexibility by enabling distillation on many sub-networks in parallel. Note also that although student neighbourhoods are distilled independently, we ultimately use them to build a bigger student model. As we want to be able to compose them properly, we constrain the student Neighbourhood $\mathcal{S}_i$ to have the same input and output dimensions as its teacher.

We demonstrate the applicability of Neighbourhood Distillation on ResNet models trained on two benchmark classification datasets: CIFAR-10 (Krizhevsky et al., 2009), and ImageNet (Deng et al., 2009). Two different settings are considered for Neighbourhood Distillation: parameter reduction and sparsification.

**Parameter Reduction**   We first explore the architecture space of ResNetV1 models by shrinking the number of parameters with a multiplier $k$, as shown in Figure 2b. On CIFAR-10, we use a ResNetV1-20 model as the teacher model where a neighbourhood is a two-layer residual block. For ImageNet, we use a ResNetV1-50 model as the teacher where the neighbourhood is a three-layer residual block. For both models, weights that are not affected by the multiplier (e.g: the first and last layer of the model) are kept as-is.

After the initial distillation step, we recompose the student neighbourhoods that have the same multiplier into a final student model. The student model is then fine-tuned on the training set using the usual Knowledge Distillation loss until convergence. Table 1 and Table 2 compare the final accuracies obtained for different student models trained with Cross-Entropy loss, Knowledge Distillation or Neighbourhood Distillation. We observe that our Neighbourhood Distillation method yields similar result as Knowledge Distillation, and both distillation methods improve on training with Cross-Entropy loss.

Table 2: Distillation results on ResNet-50 for neighbourhoods of 2 and 3 layers with different multipliers. Neighbourhood Distillation (ND) performs similarly to Knowledge Distillation (KD) and better than retraining from scratch (CE). As the gap between the student and the teacher increases, fine-tuning becomes necessary to recover good model accuracy. Our method is $3.6\times$ faster.

| MODEL($k$-NUM LAYERS) | # PARAMS | CE | KD | ND + FT | ND |
|---|---|---|---|---|---|
| RN50-1.0 | 23 M | 76.2 | - | - | - |
| RN50-0.75-2 | 21 M | 75.4 | 76.0 | **76.3** | 72.8 |
| RN50-0.75-3 | 18 M | 75.1 | 75.5 | **76.1** | 64.7 |
| RN50-0.50-2 | 17 M | 73.6 | **74.9** | **74.9** | 68.8 |
| GPU TIME (H) | - | - | 200 | 56 | 22.75 |
| SPEED-UP | - | - | $1\times$ | $3.6\times$ | $8.8\times$ |

**Sparsification** We apply Neighbourhood Distillation with the goal of distilling a sparse student from a fully-trained teacher network. During distillation, the student weights are sparsified by magnitude pruning (Zhu & Gupta, 2017) without changing the input or output shape. This allows us to consider one-layer neighbourhoods and to initialize the student weights using the teacher's learned weights. We show in Table 3 that Neighbourhood Distillation is able to reach the same performace as Knowledge Distillation for low sparsity rates, *without fine-tuning*. For high sparsity rates, the sudden drop in accuracy for models distilled with Neighbourhood Distillation is a symptom of the thresholding effect. When the target sparsity is too high, the student doesn't have enough capacity to approximate its teacher well enough.

**Timing** The main advantage of Neighbourhood Distillation is the speed-up that can be gained from training small blocks in parallel. We timed Neighbourhood Distillation and compare its runtime to that of Knowledge Distillation and report the total GPU time for each, i.e. the total time that would have been needed for one GPU P100 to run. Results are reported in Table 1, 2, and 3. Neighbourhood Distillation is $3.6\times$ faster on ImageNet and $2.3\times$ on CIFAR-10 for parameter reduction, and $1.7\times$ faster for sparsification. For parameter reduction, the end-to-end fine-tuning step is the main computational bottleneck but is faster than using Knowledge Distillation from scratch. Student neighbourhoods learn a meaningful approximation of their respective teachers, which allows the fine-tuning step to converge faster than Knowledge Distillation. On the sparsification task, while we don't fine-tune and each layer is fast to sparsify, the total number of layers (50) is what constitutes to main bottleneck when computing the total GPU time of the procedure. A further break-out of the time each phase of Neighbourhood Distillation needs is given in the appendix.

Table 3: Classification accuracies and GPU Time comparison between Neighbourhood Distillation and Knowledge Distillation. Different values of target sparsities are used for the student architecture.

| SPARSITY | ND | KD |
|---|---|---|
| 0.1 | **76.00** | 75.63 |
| 0.2 | **76.03** | 75.94 |
| 0.3 | **75.82** | 75.69 |
| 0.4 | **75.63** | 75.45 |
| 0.5 | **75.26** | 75.20 |
| 0.6 | 73.78 | **74.60** |
| 0.7 | 69.52 | **69.61** |
| 0.8 | 49.83 | **70.33** |
| 0.9 | **0.60** | 0.10 |
| GPU TIME (H) | **144** | 250 |
| SPEED-UP | $1.7\times$ | $1\times$ |

Table 4: Classification accuracies for different data-free distillation methods. Different values of $k$ are used for the student architecture including $k = 1$ (self-distillation). We compare accuracies obtained after different training methods: fully-supervised cross-entropy loss (CE), Gaussian Noise Neighbourhood Distillation (G-ND), Gaussian Noise Neighbourhood Distillation with Gaussian Noise Finetuning (G-ND+FT), Gaussian Noise Knowledge Distillation (GN-KD) and Zero-Shot Knowledge Distillation (Nayak et al., 2019) (ZSKD). Gaussian Noise Neighbourhood Distillation surpasses all other data-free distillation methods we compared to.

| $k$ | CE | G-ND | G-ND+FT | GN-KD | ZSKD |
|---|---|---|---|---|---|
| 1 | 91.60 | 91.51 | **91.51** | 16.00 | 21.50 |
| 0.9 | 91.31 | 88.78 | **88.78** | 12.97 | 22.77 |
| 0.75 | 90.89 | 80.64 | **83.74** | 13.00 | 21.54 |
| 0.5 | 90.17 | 53.40 | **54.83** | 12.74 | 22.17 |

# 5 STUDENT SEARCH

Usually, when performing Knowledge Distillation, the Student network is a variant of the Teacher network with a smaller number of parameters (Hinton et al., 2015; Romero et al., 2014; Zhang et al., 2017). Other candidate architectures with the same number of parameters could better capture the teacher's knowledge but searching for the ideal student architecture would require retraining different models from scratch and be computationally expensive. Here, we demonstrate how independently distilled neighbourhoods may be re-purposed for architecture search. Student Search, our architecture search method, uses the distilled neighbourhoods to make local decisions on a model's architectural design.

Formally, we consider $\mathcal{C}_i$ the set of possible candidates for a given neighbourhood $\mathcal{S}_i$. This set could contain variants of the same architecture with different parameters like the size of the intermediate layers, the level of sparsity, the non-linearity used. The number of possible student model architectures to distill and evaluate would be $\prod_{i=1}^{n} |\mathcal{C}_i|$. Student Search leverages Neighbourhood Distillation by first distilling all $\sum_{i=1}^{n} |\mathcal{C}_i|$ neighbourhoods completely independently. It then selects a candidate for each neighbourhood by solving a constrained optimization problem that seeks to minimize the size of the student while maximizing the quality of the selected candidates. The selected candidate neighbourhoods are then combined together to form the Student Network, which may then be fine-tuned. In practice, we find that solving the above optimization problem approximately with a greedy approach is enough to yield a good student architecture.

We demonstrate our method on the space of ResNet variants built with different multipliers. For each neighbourhood, we independently distill 10 candidates, obtained by varying the bottleneck multiplier $k$ in $\{0.1, 0.2, ..0.9, 1.0\}$. The quality of a candidate $c_i(k)$ is measured by the accuracy of the partial model $T_n \circ ...T_{i+1} \circ c_i(k) \circ T_{i-1} \circ T_1$. We solve the optimization problem greedily by selecting for each neighbourhood the smallest candidate that leads to an accuracy drop of less than $x\%$.

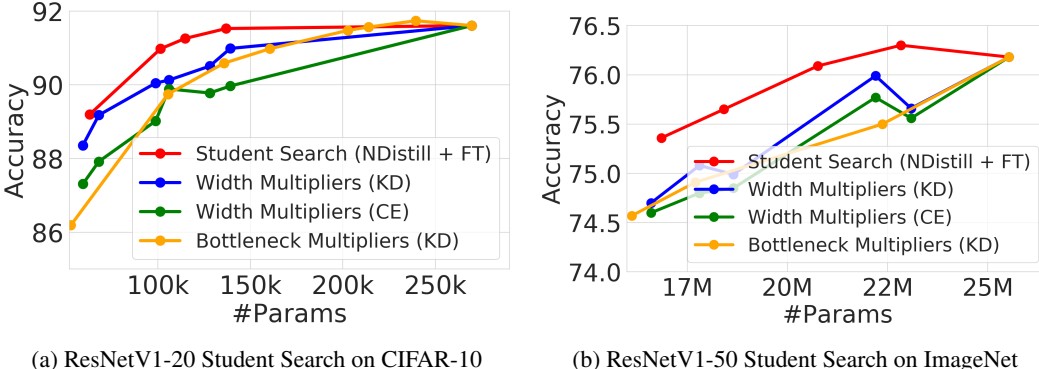

(a) ResNetV1-20 Student Search on CIFAR-10          (b) ResNetV1-50 Student Search on ImageNet

Figure 3: Trade-off curve between number of parameters and final accuracy for ResNetV1-20 and ResNetV1-50 models. Models found with Student Search are obtained by recombining units of different sizes that were distilled independently with Neighbourhood Distillation. Width multipliers refers to applying a uniform multiplier on all blocks. Bottleneck multipliers refers to applying a uniform multiplier on the inner layers of all blocks. Student Search efficiently finds better student architectures than naively applying uniform transformations to the teacher.

Figure 3a and 3b show how our Student Search compares to using a uniform multiplier on all layers when searching for a student architecture for Knowledge Distillation. Width multiplier refers to applying the same multiplier to *all* the bottleneck layers of the model. This setting cannot be explored with Neighbourhood Distillation as it modifies the input/output behavior of the neighbourhoods. Bottleneck multiplier refers to applying the same multiplier to the inner layers of each residual block: this setting is equivalent to applying Neighbourhood Distillation with the same multiplier $k$ for all neighbours. Student Search efficiently finds a better student model for distillation than uniformly applying the same transformations on the teacher model. In particular, the found stu-

dent architecture can be directly fine-tuned using the student candidates trained by Neighbourhood Distillation.

# 6 DATA-FREE KNOWLEDGE DISTILLATION

Distillation in a context where the original dataset is not accessible is of critical interest for privacy-sensitive applications. Methods for distillation in this setting essentially rely on generating synthetic image datasets, which are usually obtained by directly optimizing some input noise with regards to a pre-determined loss (Lopes et al., 2017; Nayak et al., 2019; Bhardwaj et al., 2019; Yin et al., 2020). Generating a substitute dataset using these methods is costly and have shown limited success on very deep architectures trained on complex and large-scale datasets. Previous work (Haroush et al., 2019) show that it is possible to successfully distill from Gaussian Noise in limited settings (fine-tuning or calibrating quantized models) where the gap between the student and the teacher weights is small. By switching the focus on small and shallow sub-networks, we show that Neighbourhood Distillation can be successfully adapted to use gaussian noise inputs to distill models from scratch.

To that end, we distill a ResNet-20 with bottleneck multiplier $k = \{1.0, 0.9, 0.75, 0.5\}$ using only gaussian noise as the neighbourhood' inputs. We compare our method with Zero-Shot Knowledge Distillation (Nayak et al., 2019), a data-free method that generates a synthetic training set by optimizing noise inputs with gradient descent. To highlight the benefits of training on shallow sub-networks, we also attempt to use gaussian noise inputs to distill the entire student network end-to-end. Table 4 shows that Neighbourhood Distillation outperforms both end-to-end baselines but shows degrading performance as the compression rate decreases.

These results show that distillation in a data-free regime can benefit from looking at shallow sub-networks in isolation. As deep neural networks model complex and high-dimensional functions, it is not possible to distill them successfully from unstructured gaussian noise and generally requires generating complex inputs that simulate images. On the other hand, shallow networks represent less complex functions and we have demonstrated with Neighbourhood Distillation that it is easier to approximate them using gaussian noise. Notably, when neighbourhoods are only two-layer deep like in the ResNet-20 model, the function is simple enough that it can be distilled almost perfectly using gaussian noise when $k = 1$.

# 7 DISCUSSION

In this paper, we presented an approach to distillation that breaks away from end-to-end training. We demonstrated that distilling small neighbourhoods yields many advantages compared to traditional end-to-end distillation: we free ourselves from the computational limitations that come with training very deep models and speed-up distillation by exploiting parallelism. Additionally, the trained candidates can be reused to efficiently explore an exponential number of local architecture changes with Student Search. Finally, we showed that distilling on small neighbourhoods allows us to easily distill in a no-data context by generating synthetic inputs from gaussian inputs.

We also discovered an interesting phenomenon in deep neural networks that empirically justifies the idea of Neighbourhood Distillation. The thresholding effect explains why replacing all neighbourhoods by distilled students can yield a reconstructed student with reasonable performance. It also enables efficient diagnosis for cases where the student is too small to properly approximate the teacher and is not suited for Neighbourhood Distillation.

Moving forward, understanding the thresholding effect is one key to broadening the scope of Neighbourhood Distillation. Studying what impacts the model tolerance threshold could better inform how to limit failure cases and we present in the appendix preliminary results towards that direction. More generally, our paper highlights the benefits of distilling networks in a non end-to-end manner, reaping the benefits of training on shallow networks, with the potential to revive methods such as second-order optimization methods which have been inapplicable to deep networks due to their complexity.

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

## A    EXPERIMENTAL DETAILS

**Data preprocessing**    The CIFAR-10 is standardized with per-channel train statistics and augmented at train time with random translations of 4 pixels.

**ResNetV1-20**    The teacher model is trained with standard cross-entropy loss with batch size 128 for 96k steps. We used Momentum Optimizer with a momentum of 0.9. The learning rate was increased linearly from 0.01 to 0.1 for 400 steps then decayed exponentially by a factor of 10 every 32k steps. The model is regularized with L2 weight decay of $1e-4$. The same settings were used for distillation. For Knowledge Distillation, we set $\lambda = 1.0$, and used random search on the parameter $T \in [1., 20]$ and learning rate in $[1e^{-6}, 1.]$. For Neighbourhood Distillation, we distill for 200k iterations with a learning rate of $1e^{-3}$. Finetuning is done with learning rate $1e^{-2}$, $T = 2.5$.

**ResNetV1-50**    The teacher is trained for 200 epochs with batch size 32, l2 weight decay of $1e-4$ and Momentum optimizer. The learning rate is initialized at $0.02$ and decayed exponentially with a factor $0.2$ every 30 epochs.

**Sparsification**    We use the Tensorflow Model Optimization Toolkit to sparsify our neighbourhoods. For each neighbourhood, we train for 40k steps. The sparsification rate is ramped-up to the target sparsity rate with a polynomial decay schedule for 20k steps and then maintained for another 20k steps.

## B    ADDITIONAL RESULTS

### B.1    THRESHOLDING EFFECT

We conducted an additional experiment to show how accuracy can be impacted by direct perturbation of a network's weights. For each convolution layer kernel, we choose to add noisy perturbation with varying standard deviations. We then verify whether the perturbations have an additive effect on the final accuracy by replacing weights from the original network by their perturbed version. Figure 4 shows how errors on the CIFAR-10 test set accumulate when progressively replacing the layers of a pre-trained ResNetV1-20 model, starting from the first layer. The figure compares that to the hypothetical additive regime. We see that when we constrain individual layers to cause small

Table 5: Impact of regularizing the teacher model on Neighbourhood Distillation. Accuracies are obtained before fine-tuning. Using a regularized teacher improves the accuracy of the student before fine-tuning.

| STUDENT | TEACHER | NDISTILL ACCURACY | CHANGE |
|---|---|---|---|
| RN50-0.75-3 | RN-50 | 57.63 | +0.% |
| - | RN-50+$\mathcal{N}(\sigma = 0.1)$ | 63.21 | **+5.6%** |
| RN50-0.5-3 | RN-50 | 25.7 | +0.% |
| - | RN-50+$\mathcal{N}(\sigma = 0.1)$ | 32.9 | **+7.2%** |

drops in accuracy ($< 0.1\%$), the errors accumulate sub-linearly. On the other hand, when individual layers are perturbed so much that they cause a drop of $2\%$ in accuracy each, the errors accumulate super-linearly.

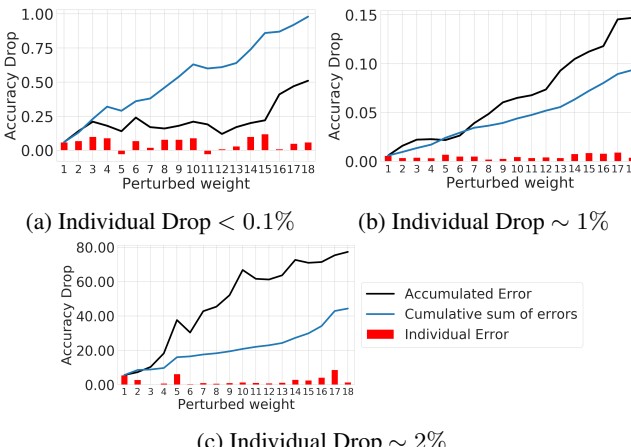

(a) Individual Drop $< 0.1\%$       (b) Individual Drop $\sim 1\%$

(c) Individual Drop $\sim 2\%$

Figure 4: Error accumulation caused by perturbing network layers. The bar plot shows the individual accuracy drop for a network with perturbed weights for one layer. The blue line plot shows the cumulative sum of errors caused by perturbing a given number of layers from left to right. The black line plot shows the empirical error accumulation evaluated on the test set after perturbing a given number of layers. (a) Individual accuracy drop of each layer is $< 0.1\%$ and the errors accumulate sub-linearly. (b) Individual accuracy drop of each layer is $\sim 1\%$ and errors accumulate linearly. (c) Individual accuracy drop of each layer is high ($\sim 2\%$) and the errors accumulate super-linearly.

We show that a simple regularization technique applied during training can increase a network's resilience to local perturbations. Considering a ResNetV1-50, we add gaussian noise drawn from $\mathcal{N}(0, \sigma)$ at train time after every residual unit. Once these models are trained, we replace each units with perturbed versions again and report the drop in accuracy. Figure 5 shows that for high noise levels $\sigma$ introduced between residual units at train time, the perturbation threshold is higher, although regularizing too much also degrades the overall performance of the original network.

More importantly, we observe that increasing the teacher's tolerance threshold directly impacts the accuracy of models trained with Neighbourhood Distillation. We take a ResNetV1-50 trained with regularization $\sigma = 0.1$ as the teacher and consider bottleneck multipliers $k = 0.5$ and $k = 0.75$ for the student neighbourhoods.

## B.2 NEIGHBOURHOOD DISTILLATION

**Look-ahead losses** In order to provide additional training signals to the student, we also consider using look-ahead losses. In addition to forcing the output of $\mathcal{S}_i$ to be close to the output of $\mathcal{T}_i$, we feed each to the next teacher Neighbourhood $\mathcal{T}_{i+1}$ and minimize the Mean Square Error of its

Thresholding effect for different perturbation amplitudes $\varepsilon$.

Figure 5: Drop in accuracy after perturbing different ResNetV1-50 models trained with Gaussian Noise $\mathcal{N}(0,\sigma)$ introduced after each unit. $\epsilon$ is the amplitude of perturbations introduced at test time. Original accuracies of the models before perturbation are given in the caption. Models trained with high gaussian noise amplitudes $\sigma$ have a higher resistance threshold to test time perturbations.

Table 6: Ablation Study. Impact of fine-tuning (FT) and look-ahead (2-LA) on final test accuracy for ResNet-20. While fine-tuned model always perform better than pre-finetuned models, the use of look-ahead tightens the gap between pre and post finetuned accuracy.

| MULTIPLIER $k$ | NDISTILL | NDISTILL + FT | 2-LA NDISTILL | 2-LA NDISTILL + FT |
|---|---|---|---|---|
| 0.90 | 91.1 | **92.0** | 91.4 | **92.1** |
| 0.75 | 89.7 | **91.9** | 90.6 | **91.8** |
| 0.50 | 85.7 | **91.3** | 87.1 | **91.5** |

output:

$$\mathcal{L}_{i,1} = \frac{1}{F_{i+2}} \mathbb{E}_{a \sim p(a)} \left[ \| \mathcal{T}_{i+1}(\mathcal{S}_i(a)) - \mathcal{T}_{i+1}(\mathcal{T}_i(a)) \|_2^2 \right] \tag{4}$$

$$a = \mathcal{R}_i(x), x \sim \mathcal{D}_{\text{train}}$$

The look-ahead loss is shown in Figure 2a. These losses can be combined to form a total loss of:

$$\mathcal{L}_i = \mathcal{L}_{i,0} + \sum_{j=1}^{n-i} \alpha_j \mathcal{L}_{i,j} \tag{5}$$

where $\alpha_j$ are hyper-parameters. While lookahead losses may improve the accuracy of the distilled model by providing additional training supervision, it also leads to increased computational cost due to needing to propagate inputs and gradients through a higher number of layers.

**Ablation Study** We perform an ablation study on the CIFAR-10 model to show how look-ahead and fine-tuning impact our Neighbourhood Distillation results. We experiment with combining two look-ahead losses, i.e $\forall j \geq 3, \alpha_j = 0$ and train each neighbourhood either without look-ahead or with two-neighbourhood lookahead. Models are evaluted before and after fine-tuning and test accuracies are reported in Table 6. We observe that training neighbourhoods with look-ahead leads to better accuracy before fine-tuning. Fine-tuned models always perform better than their non-fine-tuned counterparts. While there is no significant difference between the no look-ahead and the 2-look-ahead fine-tuned models, we also noticed that neighbourhoods distilled with look-ahead losses converge faster during the fine-tuning step - 30k iterations instead of 50k. Look-ahead losses can therefore be used to balance the cost of Neighbourhood distillation and the cost of fine-tuning.

## B.3 STUDENT SEARCH

**Motivation** We observe empirically on the ResNet-20 and ResNet-50 models that there is no reason to think that applying the same multiplier to all layers would yield the best compressed student

Table 7: Comparison of training time in GPU hours for ResNet models.

(a) Timing for ResNet20

| Method | | Training Time (h) |
|---|---|---|
| NDistill | 1-unit | 0.03 |
| | All units sequentially | 0.32 |
| | Finetuning | 1.07 |
| | All units + Finetuning | **1.39** |
| KD | | 3.2 |

(b) Timing for ResNet50

| Method | | Training Time (h) |
|---|---|---|
| NDistill | 1-unit | 1.42 |
| | All unit sequentially | 22.75 |
| | Finetuning | 33 |
| | All unit + Finetuning | **56** |
| KD | | 200 |

(a) ResNet-20 on CIFAR 10

(b) ResNet-50 on ImageNet

Figure 6: Impact of applying a multiplier $k$ on the distilled accuracy of different Residual Units. $x$-$y$ refers to unit $y$ of block $x$ of the ResNet. For each unit, accuracy remains unchanged as long as the multiplier remains above a certain threshold. This thresholding effect can be observed for different residual units, but each unit has a different threshold.

model. For a number of chosen residual units, we reduce the number of parameters by applying a multiplier $k < 1$ on the number of filters of the first layer and report the best accuracy we can get after training this variant with Neighbourhood Distillation. Figure 6 shows the final accuracy against $k$ for different units of ResNet-20 and ResNet-50. We observe another form of thresholding effect: the accuracy drops sharply below a given value of the multiplier, showing that residual units can be compressed up to a certain point before hurting the network's accuracy. Most importantly, we observe that the threshold is different for each residual unit, which suggests that a better strategy for a student architecture is to choose a different multiplier per unit.

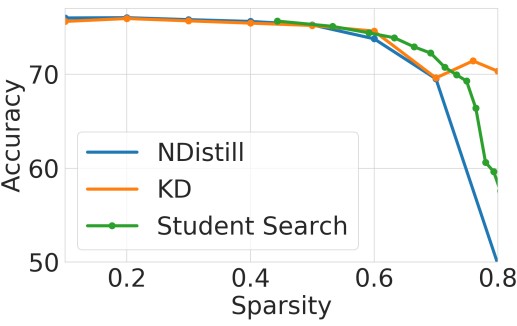

Figure 7: Student Search for Sparsification in ResNetV1-50. Models found with Student Search are obtained by recombining layers with different sparsity rates that were distilled independently with Neighbourhood Distillation. Student Search efficiently finds a better student architectures for distillation than naively applying uniform sparsity rates.

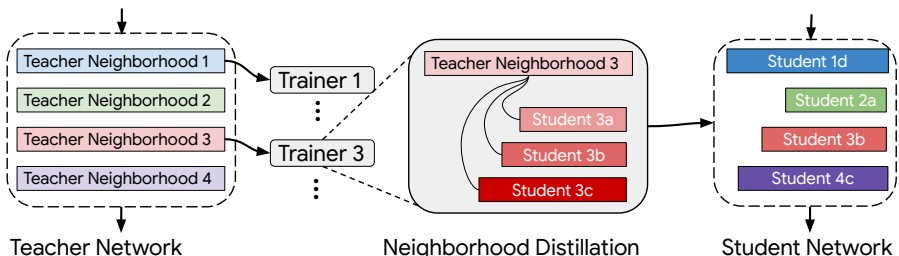

Figure 8: Student Search with Neighbourhood Distillation. A Teacher Network is broken down into several neighbourhoods. Each Neighbourhood can be used to train several student neighbourhoods independently from the rest. Selected student neighbourhoods are then merged back into a single Student Network. This allows us to explore a large search space without having to retrain all models.

