# OpenReview forum: "Neighbourhood Distillation: On the benefits of non end-to-end distillation"
_ICLR.cc/2021/Conference — Reject_

### Official Review · AnonReviewer3 · 2020-10-28
**A new training schema for KD with various applications, but inadequate experiments**

**Rating:** 5
**Confidence:** 4

**Review:**

This paper introduces Neighbourhood Distillation (ND), a new training pipeline for knowledge distillation (KD), which splits the student network into smaller neighbourhoods and trains them independently. The authors breaks away from the end-to-end paradigm in previous KD methods and provides empirical evidence to reveal feasibility and effectiveness of ND. Specially, ND can: 1) speed up convergence, 2) reuse in neural architecture search and 3) adapt to the synthetic data.

Strengths
1) The paper is well written and easy to follow. An empirical evidence of the thresholding effect is provided to explain the motivation.
2) The idea is simple and intuitive. The ND seems more like an initialization method for DNN’s local components and a finetuning procedure is sometimes needed for recovering the accuracies. Benefit from parallelism and small training components, such training schema can speed up the convergence of standard KD.
3) Several different applications are conducted to demonstrate the flexibility of ND.

Weaknesses
1) Missing a relevant paper. [1] proposes a similar blockwise knowledge distillation method. The authors should cite and explain the differences between ND and [1].
2) What is sparsification in Sec. 4? Is it the sparseness of the convolutional kernel or the channel?
3) The authors mention that the work seeks to overcome the limitations of training very deep networks. However, the ResNet50 (the deepest model in experiments) is not deep enough. Usually, it is easy to converge.
4) In Sec. 5, only the width search experiments are conducted, which is more like layer-wise or block-wise pruning. However, architecture search is a general method that can not only search the widths but also the operations. Why only mentions “This set could contain variants of the same architecture...”? Is there any limitation when the searched candidates contain different architectures/operations?
5) All the experiments are done on ResNet series. Different teacher and/or student architectures, such as VGG, ShuffleNet etc., should be considered.
6) Does the observation of thresholding effect benefit from the shortcut in Resblok？Is it suitable for plain CNN, such as VGG? Which blocks are chosen in Fig. 1(a)? Does the shallow and deep blocks have the same phenomenon when perturb small number of blocks?
7) How to record the GPU time of ND? Is it the time of paralleling on multi-GPUS or on single GPU?

I am currently leaning towards a slightly negative score but would like to see the authors' responses and other reviewer's comments.


[1] Hui Wang, Hanbin Zhao, Xi Li, Xu Tan. Progressive Blockwise Knowledge Distillation for Neural Network Acceleration. IJCAI, 2018.

---

> ### Author Response · Authors · 2020-11-13
> **Rebuttal**
>
> Thank you for review. Please find below some answers to the concerns that were raised in the review.
>
> *Missing a relevant paper. [1] proposes a similar blockwise knowledge distillation method. The authors should cite and explain the differences between ND and [1].*
>
> * Progressive Blockwise KD is indeed a relevant paper and we will add it in our related work discussion for the camera ready version. While Progressive Blockwise KD also identifies that a student network may be divided into sub-networks, we believe that it differs greatly from ND in that the training of the student is done stagewise by training one student at a time, creating a chain of students that differs increasingly from the original teacher. In Neighbourhood Distillation, **all students networks are trained in parallel and independently from each other.**
>
> *What is sparsification in Sec. 4? Is it the sparseness of the convolutional kernel or the channel?*
>
> * Sparsification refers to **convolutional kernel sparsity**. We mention this in the sparsification paragraph p6.
>
> *Is there any limitation when the searched candidates contain different architectures/operations?*
>
> * We would consider different operations or architectures to be valid for the candidates as well. We currently limit our candidates to have the **same input and output shapes as the teacher** they are trained with, although we believe that this limitation could be bypassed by using a trained encoder/decoder network similar to the Hint-training approach.
>
> *Which blocks are chosen in Fig. 1(a)? Does the shallow and deep blocks have the same phenomenon when perturb small number of blocks?*
>
> * Figure 1.a shows the thresholding effects for a ResNet-20 that has 9 residual blocks. Each line corresponds to a different number of perturbed blocks starting from the base of the network. So 1 block corresponds to the first block, 2 blocks to the first two blocks, etc.
> The depth of the blocks seem to also impact the phenomenon, which is why we compare ResNet-20 and EfficientNet blocks (as they have different depths).
>
> *How to record the GPU time of ND? Is it the time of paralleling on multi-GPUS or on single GPU?*
>
> * GPU time is computed by **multiplying the total wall-time of running the distillation by the number of GPUs that were used**: effectively giving a measure of the training time needed if the user only has access to one GPU at a time. As such, for the Neighbourhood Distillation column, the reported speed-up doesn’t take into account the possibility of training all neighbourhoods in parallel. This means that when training with multiple GPUs, the wall time speedup one gets would be higher than what we report.

---

### Official Review · AnonReviewer4 · 2020-10-29
**Official Blind Review #4**

**Rating:** 4
**Confidence:** 4

**Review:**

I vote for rejection at this moment. This paper proposes a new knowledge distillation method called Neighbourhood Distillation to speed up knowledge distillation process. Compare to conventional KD approaches, this paper can achieve 3-4 times speed and obtain similar performance. Experiment on several benchmarks shows the proposed method can be beneficial in several application.
 Overall, I think the idea to speed up the process of knowledge distillation is a direction that could be discussed in the future,  while the training cost of current KD in experiments in this paper is almost affordable and the speedup is not very impressive. Besides, several concerns need to be addressed.

####Pros:
1) The observation of  Thresholding effect is somewhat novel to me.
2) The methodology part is clear and easy to understand.
3) This paper has investigated the ND in several tasks, including NAS, data-free KD,  and this is worth to mention.

#### Cons:
1)  For me, the connection in Thresholding Effect and NEIGHBOURHOOD DISTILLATION is a little weak. I noticed the author claim this motivated to use NEIGHBOURHOOD DISTILLATION: each student neighbourhood is trained to match its output of teacher. Such an idea is already used similarly way in Hint-Training and the following work.
2)  The goal of this paper is to speed up the KD process, however existing cost of KD experiments in the paper is affordable. And the speed up is not very impressive.
3)   The paper should add discussion some recently KD methods, especially matching intermediate features between student and teachers.
4)  The results of Table 1 should be run several times and report the average mean.
5)  The GPU time (H) is not a very objective index, could you provide the flops of each method?

---

> ### Author Response · Authors · 2020-11-13
> **Rebuttal**
>
> Thank you for your review. We'd first like to answer some questions related to the method and paper.
>
> *For me, the connection in Thresholding Effect and NEIGHBOURHOOD DISTILLATION is a little weak. I noticed the author claim this motivated to use NEIGHBOURHOOD DISTILLATION: each student neighbourhood is trained to match its output of teacher. Such an idea is already used similarly way in Hint-Training and the following work.*
>
> * While bearing some similarities with Hint training in that we distill using activation outputs matching, our work has a distinguishing feature in that the **student neighbourhoods are trained independently from each other** (i.e: they are not aware of each other during the distillation process). In Hint training, the training of one student subnetwork will depend on the activation output of the preceding student subnetwork: this creates a sequential dependence of the subnetwork on all preceding student sub-networks. Our method in comparison is not limited by these constraints because Neighbourhood Distillation trains each student independently by using the activation output of the preceding teacher subnetwork as inputs (as shown in Figure 1).
> * Since each student neighbourhood is trained without being aware of the other students, a reasonable concern is that individual errors of each student will accumulate when they are recomposed together. A study of the thresholding effect in ResNets and EfficientNets shows that this concern is not important as long as we ensure that these individual errors remain below a certain threshold.
>
> *The goal of this paper is to speed up the KD process, however existing cost of KD experiments in the paper is affordable. And the speed up is not very impressive.*
>
> * We disagree with the assessment that the goal of the paper is solely to speed up the KD process. The goal of the paper is to present Neighbourhood Distillation as **a general new distillation method  that breaks away from the end-to-end training paradigm**. In the paper, we not only explore possible speed-up gain but also show how Neighbourhood Distillation can be used to support NAS methods and Data-Free Distillation methods. This is reiterated in the abstract and the contributions summary.
>
> *The paper should add discussion some recently KD methods, especially matching intermediate features between student and teachers.*
>
> * We mention several KD methods in the Contributions section related to Knowledge Methods, in particular matching intermediate features and Hint Training. As stated, we believe our work differs from previous methods in that it **departs from the end-to-end sequential training paradigms** that these methods adopt. We will add a more detailed discussion on what these methods do and how they differ from ours in the camera-ready version.

---

### Official Review · AnonReviewer2 · 2020-11-02
**An interesting work, but lacking details**

**Rating:** 5
**Confidence:** 4

**Review:**

This paper studies knowledge distillation in the context of parallelly training sub-networks (called neighbourhoods) instead of commonly used end-to-end training paradigm. The authors explore the applications of the proposed neighbourhoods distillation in improving sparse networks,  searching a good student structure given the teacher and knowledge distillation merely using synthetic data. Both CIFAR and ImageNet datasets are considered in the experiments.

The proposed method is interesting. My main concerns are as follows.

--- Thresholding effect.

The experimental findings regarding thresholding effect is interesting. In common understanding of CNNs, non-linear activations play a critical role in training. When perturbing a CNN or its  neighbourhoods by artificially introducing some gaussian noise of amplitude epsilon to each activation output, it will break non-negative property. What is the underlying reason why it does not affect final predication once the gaussian noise of amplitude epsilon is small enough?  Why the behavior for injecting noise to network weights is different from noise injection to activations? They are not clear enough. Besides, more details on how to inject noises to a network or its neighbourhoods would be useful.

--- Neighbourhood distillation (ND).

In the formulation, the authors assume the teacher is pre-trained and available. And the authors claim that ND can be performed in a much faster manner, compared to conventional KD. However, to my understanding, it is not clear enough.

a) From Figure 2, I have not seen obvious differences against conventional KD or intermediate feature/attention maps guided KD, if we consider a subnetwork as the student.

b) The authors claim "These activation maps can be pre-computed and stored before training. This makes each student neighbourhood extremely fast to train, as one does not need to compute activations through a big model in each training step." This can also be used to conventional KD or intermediate feature/attention maps guided KD, if we consider a sub-network as the student.

c) The authors claim "Different student neighbourhoods can be distilled independently from each other and composed into
a final student model" and then the final model is fine-tuned. How about computational resource cost when training them parallelly? Does the speed-up is also benefited from parallel training?

d) In the equations such (1) and (3), many notations or terms are not clarified.

I am confused by those points, which makes me think the presentation and applications of ND are not convincing.

--- Experiments

In the experiments, the authors assume the student and teacher networks have strong relations, multiplier k. However, convention KD and its variants do not have any structural constraints between the student and teacher networks. That is, they usually have different structures, such as DenseNets vs. ResNets.

KD methods are not limited to two-stage training, as many KD variants enable on the fly training of teacher/peer together with the student. Is it possible to apply ND to such kind of training scenarios?

Regarding speed-up results over KD, it lacks details how they are implemented/measured.

---

> ### Author Response · Authors · 2020-11-13
> **Rebuttal**
>
> Thank you for your review. Below are some clarifications to questions that were raised.
>
> *When perturbing a CNN or its neighbourhoods by artificially introducing some gaussian noise of amplitude epsilon to each activation output, it will break non-negative property. What is the underlying reason why it does not affect final predication once the gaussian noise of amplitude epsilon is small enough?*
>
> * All experiments consider **pre-ReLU activations** and not post-ReLU. This will be clarified in the camera-ready version.
>
> *Why the behavior for injecting noise to network weights is different from noise injection to activations?*
>
> * We would argue that the behaviour is not different. We show in Figure 1 the behaviour of a network after injecting noise into the activations while Figure 4 in the appendix shows the behaviour of a network after injecting noise in the network weights. Both figures show different manifestations of the same effect: when perturbations are small enough, they don’t accumulate to degrade the network’s performance.
>
> *a) From Figure 2, I have not seen obvious differences against conventional KD or intermediate feature/attention maps guided KD, if we consider a subnetwork as the student.*
>
> * Figure 2 shows the computational graph for training one sub-network as the student. The figure should highlight two key differences with conventional KD or intermediate maps feature guided KD. First, our student subnetwork is not limited to a “prefix” network as is done in Hint-training methods. Second, in Neighbourhood Distillation, **students are not trained sequentially**. In Hint training-like methods, the intermediate supervision is used to train the module sequentially or end-to-end. In Neighbourhood Distillation, students are trained independently from each other and in parallel. This is further clarified in the method description (Section 4, page 5 above equation 3).
>
> *b) The authors claim "These activation maps can be pre-computed and stored before training. This makes each student neighbourhood extremely fast to train, as one does not need to compute activations through a big model in each training step." This can also be used to conventional KD or intermediate feature/attention maps guided KD, if we consider a sub-network as the student.*
>
> *  In conventional KD or intermediate feature guided KD, the training of one student sub-network will depend on the activation output of the preceding student sub-network. This sequential dependence limits the efficiency of precomputing activations since they would either need to be regularly updated to match the updates of the preceding student subnetwork or be computed after having completed the training of the preceding student subnetwork. Our method in comparison is not limited by these constraints because Neighbourhood Distillation trains each student **independently by using the activation output of the preceding teacher subnetwork** as inputs.
>
> *c) The authors claim "Different student neighbourhoods can be distilled independently from each other and composed into a final student model" and then the final model is fine-tuned. How about computational resource cost when training them parallelly? Does the speed-up is also benefited from parallel training?*
>
> * Table 1 and Table 2 both show the computational cost in GPU hours of distillation using either Neighbourhood Distillation and Knowledge Distillation. **GPU hour is computed by multiplying the total wall-time of running the distillation by the number of GPUs that were used**: effectively giving a measure of the training time needed if the user only had access to one GPU at a time.
> * **NDistill also scales up linearly with the number of GPUs**. One can distill a ResNet-1000 with 1000 GPUs in parallel and it would take the same wall time as training a ResNet-50 with 50 GPUs. The scale-up doesn’t happen with regular Knowledge Distillation because: 1. The asynchronous nature of training with multiple GPUs means that there are diminishing returns on adding more GPUs, and in practice one doesn't see any speed improvement by using more than 50 GPUs. 2. When training synchronously, one needs to co-locate the GPUs which again limits the parallelism.
>
> *d) In the equations such (1) and (3), many notations or terms are not clarified.*
>
> * We will clarify in the text:
> For equation 1: $F_i$ is the flattened feature dimension of activation at layer i.
> Fo equation 3: $x$ is the image input of a network, $y$ is the real label, $\tau$ is the temperature used to soften the logit outputs as is commonly used for Knowledge Distillation.
> The notation $\hat(y)^S_\tau$ is defined in the second line of equation 3. It's a shortcut for the predictions obtained from network $S$ after smoothing the logits with temperature $\tau$.
> $\cal{S}$ and $\cal{T}$ are defined in the text that precedes the equation and refer to the student and teacher network respectively

---

### Decision · Program_Chairs · 2021-01-07
**Final Decision**

**Decision:**

Reject

**Comment:**

The paper proposes a layer-wise or block-wise distillation scheme, Neighbourhood Distillation, that aims to reduce the training time and to improve parallelism when distilling large teacher networks. By breaking down the end-to-end distillation objective into blocks, the proposed method enables faster distillation when applied to model compression and block-wise architecture search. Several concerns of reviewers were addressed during the rebuttal period by the authors.

However, there are still some concerns among the reviewers after the discussion:

1) Computational cost-benefit of the proposed KD method seems marginal in comparison to the baseline, given that we still have to train all the neighborhoods in parallel and potentially to fine-tune in the end.

2) It was brought up by a couple of the reviewers that the experiments lack diversity. It would be great to a clearly defined metric applied to a wide variety of the model architectures and datasets. It will also strengthen the paper by providing more details on the experiments.

The basic idea is interesting, but the paper needs further development and modification for publishing.